# Global Epidemiology of Invasive Infections by Uncommon *Candida* Species: A Systematic Review

**DOI:** 10.3390/jof10080558

**Published:** 2024-08-07

**Authors:** Sandra Pinho, Isabel M. Miranda, Sofia Costa-de-Oliveira

**Affiliations:** 1Faculty of Medicine, University of Porto, 4200-319 Porto, Portugal; sdrpinho@gmail.com; 2Cardiovascular R&D Centre UnIC@RISE, Department of Surgery and Physiology, Faculty of Medicine, University of Porto, 4200-319 Porto, Portugal; imiranda@med.up.pt; 3Division of Microbiology, Department of Pathology, Faculty of Medicine, University of Porto, 4200-319 Porto, Portugal; 4Center for Health Technology and Services Research—CINTESIS@RISE, Faculty of Medicine, University of Porto, 4200-319 Porto, Portugal

**Keywords:** invasive infection, *Candida* spp., epidemiology, uncommon yeast, *Candida auris*

## Abstract

Emerging and uncommon *Candida* species have been reported as an increasing cause of invasive *Candida* infections (ICI). We aim to systematize the global epidemiology associated with emergent uncommon *Candida* species responsible for invasive infections in adult patients. A systematic review (from 1 January 2001 to 28 February 2023) regarding epidemiological, clinical, and microbiological data associated to invasive *Candida* infections by uncommon *Candida* spp. were collected. In total, 1567 publications were identified, and 36 were selected according to inclusion criteria (45 cases). The chosen studies covered: *C. auris* (*n* = 21), *C. haemulonii* (*n* = 6), *C. fermentati* (*n* = 4), *C. kefyr* (*n* = 4), *C. norvegensis* (*n* = 3), *C. nivariensis* (*n* = 3), *C. bracarensis* (*n* = 1), *C. duobushaemulonii* (*n* = 1), *C. blankii* (*n* = 1), and *C. khanbhai* (*n* = 1). Over the recent years, there has been an increase in the number of invasive infections caused by uncommon *Candida* spp. Asia and Europe are the continents with the most reported cases. The challenges in strain identification and antifungal susceptibility interpretation were significant. The absence of clinical breakpoints for the susceptibility profile determination for uncommon *Candida* spp. makes interpretation and treatment options a clinical challenge. It is crucial that we focus on new and accessible microbiology techniques to make fast and accurate diagnostics and treatments.

## 1. Introduction

Over the years, the enormous advances in science, including medicine, have allowed a remarkable increase in average life expectancy. However, the rise in longevity has been partly at the expense of better therapeutic options [1]. Intensive care medicine, oncology, and intensive care units are some areas in which prophylaxis against bacteria and fungi has played a lead in preventing opportunistic infections in immunosuppressed individuals [2,3]. In contrast, the increased use of broad-spectrum antibiotics has contributed to increased opportunistic infections, particularly (but not exclusively) in immunocompromised patients, especially fungal infections [4,5,6,7].

The clinical manifestations of fungal infections include a broad spectrum ranging from superficial lesions to life-threatening invasive infections. Since the early 1980s, fungi have been leading in the etiology of invasive infections, with high mortality and morbidity rates. Over the last decade, there has been a dramatic increase in scientific studies associating sepsis with prior fungal infection [3,4,8,9]. According to *Bassetti* and his colleagues, invasive infections in intensive care units are increasing, and 80% of these infections are caused by *Candida* species [9]. *Candida* spp. are commonly distributed in the environment and are part of our normal microflora [10]. A transition from commensal to a pathogenic agent and evolution for invasive infections occur primarily in immunodepression situations or when barrier leakage occurs [11]. *Candida* infections can be classified as cutaneous, mucosal, or invasive when the infection reaches the bloodstream or other typically sterile sites and deep tissues [12]. *Candida albicans* is the most frequent species that causes invasive infections with severe repercussions on patient prognosis, morbidity, and mortality and the economic health system [10]. However, in recent decades, we have seen an increase in non-*albicans* invasive *Candida* infection (ICI) reports worldwide [13,14,15]. A recent ten-year analysis of the distribution of these species indicated that *C. glabrata* (new nomenclature: *Nakaseomyces glabrata*) is the most common, followed by *C. parapsilosis, C. tropicalis, and C. krusei* (new nomenclature: *Pichia kudriavzevii*) [12,16,17].

Uncommon *Candida* species play a significant role in the landscape of candidemia due to their emerging resistance to antifungal treatments, diagnostic challenges, varied clinical manifestations, epidemiological shifts, impact on patient outcomes, and influence on treatment guidelines. In 1991, the first case of *C. haemulonii* was reported in the USA and associated with a poor prognosis [18]. In 2009, *C. auris* was identified for the first time [19,20]. Since then, numerous studies have reported cases of invasive infections associated with high mortality and morbidity rates. These emerging pathogens have a low susceptibility to antifungal drugs and affect mainly immunocompromised patients [21]. *C. nivariensis* and *C. bracarensis* have been associated to distinct antifungal susceptibility pattern, and, together with *C. glabrata (sensu stricto),* form the *C. glabrata* species complex [22]. The unambiguous identification of *C. nivariensis* and *C. bracarensis* through traditional methods has been reported [22]. *C. fermentati* belong to the *C. guilliermondii* species complex, a heterogeneous taxonomic group with several morphologically indistinguishable species [23,24]. Consequently, ICI by uncommon *Candida* spp. has led to a public health concern worldwide such as *C. auris* [25,26,27].

This work aimed to systematically review the worldwide literature regarding the epidemiology, diagnostics, susceptibility profile, and clinical outcome of uncommon *Candida* species responsible for invasive infection.

## 2. Materials and Methods

The systematic review was carried out following the Cochrane Handbook for Systematic Reviews of Interventions and reported following the Preferred Reporting Items for Systematic Review and Meta-Analyses (PRISMA) guidelines [28,29].

### 2.1. Defining the Problem and Establishing the Guiding Question

The PICO anagram (population, intervention, comparison, and outcome) was defined and represented in Appendix A to perform the guiding question. According to the guiding question, this work did not use the comparison element [30]. The guiding question of this research was: “What is the clinical information, diagnostics, susceptibility profile, and clinical outcome of patients with invasive infections by uncommon *Candida* species?”

### 2.2. Data Source and Search Strategy

An electronic literature search comprising terms such as [(*Candida* OR “*Candida* species” OR “*Candida* spp”) AND (“invasive” OR “deep-seated” OR “bloodstream infection”) AND (emergent OR emergence OR emerging) AND (resistance OR resistant OR “multiple drug resistance” OR “multiple drug resistant” OR Pan OR “decrease susceptibility” OR “treatment failure” OR “poor outcome”)] was carried out in databases Pubmed, Web of Science, and SCOPUS to identify case reports of patients with invasive infection caused by uncommon *Candida* species. The study was conducted from 1 January 2001 to 28 February 2023. The search included all publication types except reviews or systematic reviews, and no language restrictions were applied.

### 2.3. Eligibility Criteria

Studies were included in this review if they met the following criteria: (1) information reporting invasive infection by uncommon *Candida* spp in patients (≥18 years old), (2) predisposing factors/underlying medical conditions, (3) method(s) of diagnosis, and susceptibility testing results, (4) information regarding treatment and other patient management strategies, and (5) patient outcome. Uncommon *Candida* species were defined as all *Candida* spp. except *C. albicans*, *C. glabrata*, *C. parapsilosis*, *C. tropicalis*, *C. lusitaniae* (new nomenclature: *Clavispora lusitaniae*), *C. guilliermondii* (new nomenclature: *Meyerozyma guilliermondii*), *C. krusei,* and *C. dubliniensis*. Invasive *Candida* infection, also known as invasive candidiasis or systematic candidiasis, involves bloodstream and/or deep-seated infections such as those in the brain, kidney, heart, liver, and lungs [12].

The following exclusion criteria were applied: (1) studies regarding noninvasive fungal infection, (2) studies regarding infections by *C. albicans*, *C. glabrata*, *C. parapsilosis*, *C. tropicalis*, *C. lusitaniae*, *C. guilliermondii*, *C. krusei,* and *C. dubliniensis,* and (3) studies with no patient clinical information and outcome, and studies with no laboratory data.

### 2.4. Data Extraction and Synthesis

The screening of publications was carried out by two independent reviewers (SP and SCO) based on the eligibility criteria. The literature screening was based on a 2-step process. First, the extracted studies were uploaded to EndNote 20.4.1 and Rayyan software for duplicate removal and further selection according to the title and abstracts [31]. The second phase was the review of the full text of selected studies. Disagreements were resolved by consensus. The PRISMA flowchart (Appendix A) summarizes the literature search and screening strategies. A protocol was defined to independently synthesize and compare the extracted data from the selected studies. Data extracted included information about the publication date, country, age, gender, clinical history/underlying medical conditions, type of sample, *Candida* identification methods, susceptibility testing method and results, clinical outcome, microbiology identification method, and antifungal treatment or other management strategies.

### 2.5. Risk of Bias (ROB) Assessment

To evaluate the risk of bias in the studies included in this review, we used the National Institute of Health (NIH) Quality Assessment Tool for Case Series Studies. A 3-point scale was used to grade nine questions to evaluate the potential source of bias as good, fair, or poor. No studies were excluded based on quality. ROB assessment was performed independently by SP and IMM.

## 3. Results

### 3.1. Publication Characteristics

Detailed data on scientific papers included in this review are provided in Appendix A. We identified 1567 publications, including 440 from PubMed, 550 from Web of Science, 577 from SCOPUS, and 47 publications, which were later manually added from the reference lists of identified papers. After checking for duplicates, 916 studies were enrolled for the title and abstract screening; 825 were excluded, and 88 were enrolled for a full-text screening and assessment. 

Thirty-six publications reporting on 45 patients were included. Cases from Asia were the most common (*n* = 22, 47.8%), followed by Europe (*n* = 15, 33.3%) and America (*n* = 8, 17.8%). No data were obtained from African countries and New Zealand (Figure 1). The number of included cases per year is represented in Appendix A.

Asia

Asia was the continent with the highest number of included cases in the present work, with a total of 22 cases. *C. auris* was the most isolated agent (*n* = 10), all related to candidemia (Appendix A). A case of *C. auris* candidemia was reported in Taiwan, related to a 64-year-old male Taiwanese patient who has lived in Vietnam for the past five years [32]. The isolates were closely related to the South Asian Clade (I). In the Middle East, two isolates of *C. auris* related to candidemia reported in Oman in 2017 were nested into two distinct clusters: the Indian and UK clusters [33]. Both patients never traveled outside their country.

*C. fermentati* was the second most reported agent associated with invasive infection, with four cases, including one case in which *C. famata* was also identified in the same isolate [34]. Two cases of *C. haemulonii* candidemia were also reported in Taiwan and South Korea, respectively, in 2010 and 2011 (Figure 2A and Appendix A) [32,35]. *C. kefyr*, *C. nivariensis*, *C. norvegensis*, and *C. duobushaemulonii* were documented in one case each, all related to candidemia (Appendix A).

In 2022, a new *Candida* spp. associated with an invasive infection case was reported for the first time in 2022: *C. khanbhai* [36]. A 55-year-old man with no known comorbidities was hospitalized in 2014 for the treatment of new-onset hospital-acquired pneumonia and ended up succumbing to *C. khanbhai* candidemia.

Europe

It is worth noting the incidence of emerging infections in Europe in the last decade. This review included 15 cases of invasive infection in Europe reported from distinct European countries: six cases in Spain [37,38,39], three in Greece [40,41], two in France [42,43], and one in Belgium [44], Netherlands [45], Switzerland [46], and Italy [47] (Appendix A and Figure 2B). Candidemia was the most documented invasive infection, although hepatic infection and intraabdominal abscesses were also reported [47,48] (Appendix A).

*C. auris* was the most reported agent (*n* = 9), and all cases were associated with candidemia, with the exception of one related to a hepatic abscess in Greece [48]. Three cases of *C. auris* invasive infection had Clade information [41,44,45]. Two isolates of. *C. auris* related to candidemia cases were reported in Belgium and the Netherlands and were associated with Clade I. Both patients had a recent history of hospitalization in India and Kuwait, respectively. Another case of *C. auris* invasive infection was documented in 2023 in Greece [41]. The isolate was also related to Clade I. However, in these cases, the patient had no history of traveling outside of their country. The authors suggested a possible horizontal transmission.

*C. norvegensis* and *C. nivariensis* were documented in Spain and Italy [38,47]. Both were transplanted liver patients, and *C. norvegensis* was isolated from blood and bile specimens, respectively.

America

A total of eight cases reported in the American continent were included: three cases of *C. haemulonii* [49,50,51], two cases of *C. auris* candidemia, and one case of *C. kefyr*, *C. blankii* [52], and *C. bracarensis* (Appendix A and Figure 2B) [53]. *C. haemulonii* was reported in South America; in contrast, North America had distinct uncommon *Candida* spp. in the USA and Canada (Figure 2C). Most cases were associated with candidemia (Appendix A). A case of liver abscess associated with *C. haemulonii* was reported in Peru in 2021 [51]: a 72-year-old woman with congenital polycystic kidney liver disease in end-stage renal disease was treated with caspofungin with a full recovery.

A recent *C. blankii* case was reported in the USA (2021): candidemia with possible endocarditis in a 63-year-old man who was immunocompetent but with a recent hospitalization history due to sepsis [52]. Another uncommon invasive infection was documented, also in the USA, related to a 64-year-old man with a malignancy pathology and a history of recent use of broad-spectrum antibiotics, in which *C. kefyr* was isolated in pleural fluid.

### 3.2. Risk of Bias—Quality Assessment

Three studies were rated as Fair due to a lack of information regarding the study question [54], study population [55], and the type of intervention [56] (Appendix A).

### 3.3. Patient Characteristics

Among the 45 patients, 12 (median 26.7%) were female, and 33 cases (median 73.3%) were male patients. In Europe, the male proportion was significantly higher in comparison with females (male: 80%; female: 20%). In contrast, the Middle East had a major proportion of female patients (female: 80%; male: 20%). The age range was from 26 to 88 years, with a mean and standard deviation of 59.1 ± 16.6 years.

### 3.4. Types of Infection

The most common type of infection was candidemia, with 40 cases in a total of 45 cases (median: 88.9%). In general, *C. auris* was the most frequently reported yeast to cause a bloodstream infection (*n* = 20, median 52.5%). *C. haemulonii* was presented in five cases of candidemia (12.5%); *C. fermentati* was isolated from four fungemia cases (10%); and *C. famata* was also isolated in one of these patients [34]. *C. norvegensis* and *C. nivariensis* were responsible for each one of the three cases of candidemia. One case had an isolation of uncommon *Candida* in blood, but no symptoms were associated with candidemia [45].

*C. blankii*, *C. duobushaemulonii*, *C. bracarensis*, and *C. khanbhai* invasive infections were less common, with only one case of each one of these [36,52,53,56]. However, atypical invasive infections were most common with these agents: endocarditis was associated with a *C. blankii* invasive infection and *C. kefyr* was reported in a case of pyelonephritis [40,52].

Eighteen patients (40%) were reported to have more than one *Candida* agent infection (bacterial and fungal). These cases were reported to have gastrointestinal, urinary, and upper respiratory tract infections or other unspecified infections at the same time or recently.

### 3.5. Underlying Conditions, Hospitalization Time, Treatment, and Clinical Outcome

Malignancy diseases were the most common underlying condition, present in 16 cases among a total of 45 cases. Diabetes mellitus was noted in 8 patients [32,33,41,52,56,57,58], and transplantation was documented in 10 cases [34,38,46,47,48,53,58,59]. Hospitalization time information was absent in 6 cases [45,46,50,52,55,57,58,60]; however, the remaining 42 cases had a minimum hospitalization time of 8 days and a maximum of more than 100 days. Only 21 cases had information related to mechanical ventilation: 17 patients needed intubation/ventilation. In 34 cases, prior exposure to antibiotics belonging to the broth spectrum was recorded; in 4 cases, this information was not provided. The antifungal agents used for the treatment of invasive candidemia were also documented in Appendix A. In a total of 45 patients, 5 cases had no information related to antifungal treatment [49,50,56,61,62].

A patient with *C. khanbhai* isolated in blood culture died before starting treatment [36]. Fluconazole and amphotericin were the most common antifungals used, documented in 14 cases. Caspofungin was the most documented echinocandin (*n* = 11), followed by anidulafungin and micafungin, 9 and 8 cases, respectively (Appendix A). Fluconazole was also the antifungal agent most used in prophylaxis and empirical treatment (Appendix A).

### 3.6. Mortality

In the 45 cases of uncommon *Candida* spp. invasive infection in our review, 19 patients died (42.2%). However, one death was not attributed to emergent *Candida* spp. invasive infection [53].

The average age of the population who ended up dying was 59.4 ± 16.6 years (range of 26–88 years). According to the gender category, 8 of 33 male patients and 7 of 12 women died (24.2% and 58.3%, respectively). The mortality specified for each uncommon *Candida* spp. is represented in Appendix A.

In a total of 21 cases, 11 patients with *C. auris* infection died (52.3%). Two of the three *C. fermentati* cases and half of the *C. norvegensis* patients succumbed (66.7% and 50%, respectively).

There is only one case of *C. khanbhai*, in which the patient had an unfavorable clinical outcome [36]. In the case of *C. duobushaemulonii* candidemia, with unfavorable clinical outcomes, the treatment for *Candida* invasive infection was withdrawn due to poor neurological recovery [56]. No death was documented in the four patients with *C. kefyr*, and the three cases of *C. nivariensis* invasive infection were included in this study.

### 3.7. Laboratorial Diagnosis

This review documented 45 cases of uncommon Candida species as causative agents of invasive infection. Most of the authors reported that the identification of uncommon species was not initially correct. One case mentioned that the samples went to an external laboratory without further information [52]. Appendix A represents the misidentification isolates presented in our included cases. Among the 44 cases with information regardless of identification methods, only 11 cases had a corrected *Candida* identification in the first attempt (25%) through the VITEK^®^, API^®^ or MicroScan^®^ system. In 24 cases (54.5%), the infectious agent was misidentified for the first time or not clearly identified with conventional methods.

Confirmation was performed through MALDI-TOF and, in some cases also, PCR sequencing (ITS and D1/D2 domain). It is important to mention that misidentification was also performed with MALDI-TOF: one case of *C. auris* candidemia was initially identified as *C. haemulonii,* and the identification of *C. kefyr* and *C. duoshaemulonii* candidemia (one case of each) failed [46,57].

The VITEK^®^ system was one of the most used identification routine techniques in 19 of 44 cases (43.2%). However, *C. auris* misidentification was common, with 11 in a total of 21 isolates with the VITEK^®^ system. *C. haemulonii* was the most frequent agent identified as *C. auris* (*n* = 4).

Fourteen cases were initially identified through the API^®^ system, and five cases with the MicroScan system. However, misidentification was presented in 11 and 4 cases with the API^®^ system and MicroScan^®^ system, respectively. Thirteen cases used identification agents with only one method, and DNA sequencing was the most used in these cases (*n* = 5).

The susceptibility tests of the reported cases included in the present work are summarized in Appendix A. Information related to the susceptibility determination method(s) was absent in four articles [52,59,63,64]. To evaluate the susceptibility of the isolated agents, a variety of techniques were used, such as the Clinical and Laboratory Standards Institute (CLSI) and the European Committee on Antimicrobial Susceptibility Testing (EUCAST) broth microdilution, CLSI microdilution, Etest^®^, Sensititre YeastOne^®^, and automated VITEK 2^®^ yeast susceptibility system. CLSI broth microdilution was the most used method (*n* = 16). EUCAST broth microdilution method was used exclusively in European reported cases (*n* = 6), including in three isolates whose susceptibility profiles were also analyzed with another technique, the broth microdilution EUCAST method compared with the Etest^®^ technique in *C. nivariensis* and *C. kefyr* isolates and also compared with the Sensititre YeastOne^®^ technique in the *C. auris* isolate [40,42,44].

In one case related to a *C. kefyr* invasive infection in 2004, CLSI broth microdilution was performed [40]. The automated VITEK2^®^ yeast susceptibility system was applied only in on case reported in 2014, in Italy [47]. Another distinct colorimetric microdilution antifungal susceptibility test, MICRONAUT, was recently performed in one fatal case of *C. auris* [58].

Different mutations in *FKS*1 and *ERG*11 genes, associated with antifungal drug resistance, were analyzed in four and two studies, respectively (Appendix A) [32,58,59,65]. *FKS*1 was presented in two *C. auris* invasive infection cases, and one *C. fermentati*on candidemia and *ERG*11 was detected in one immunocompromised patient with a *C. auris* fatal infection. A case reported by Tsai, M. H. et al. related to *C. auris* candidemia also evaluated *FKS*1 and *ERG*11 mutations, but none of them were detected [32].

#### 3.7.1. *C. auris*

Regarding the susceptibility profile, *C. auris* had, in general, high values for fluconazole with a range between 2 and ≥256 mg/L; 14 of 21 *C. auris* isolates had a high MIC to fluconazole, according to the Center for Disease Control (CDC) breakpoints [66]. One case did not have information regarding fluconazole [64]. All the nine isolates of *C. auris* reported in Europe had high MICs of fluconazole (Appendix A).

Susceptibility testing for amphotericin B was performed in all cases, and the results revealed that 4 of 21 *C. auris* isolates had high MICs to this antifungal, according to the CDC breakpoints [66]. Curiously, three of the four *C. auris* cases with a high MIC to amphotericin B also showed high values for fluconazole [33,41,67].

The susceptibility testing results for other azoles showed 10 *C. auris* isolates with a high MIC to voriconazole, 2 cases of posaconazole, and 1 case of itraconazole (Appendix A). One isolate documented in Russia in 2019 exhibited a high MIC to caspofungin [55]. Most of the studies reported low MIC values to anidulafungin and micafungin.

A particular *C. auris* isolate reported in the USA presented an *FKS*1 mutation and revealed, according to the authors, high MICs to echinocandins [65] (Appendix A). Another isolate of *C. auris* associated with invasive infection exhibited *FKS*1 and *ERG*11 mutations and also high MICs to echinocandins and fluconazole [58]. All *FKS*1 mutations occurred in hotspot 1.

#### 3.7.2. *C. haemulonii*

One reported case of *C. haemulonii* had no information related to the susceptibility testing method used [63]. In this case, the authors mention that the isolate is susceptible to echinocandins such as caspofungin or micafungin, and it was not resistant to new triazoles (e.g., voriconazole).

The cutoff of CLSI was used in four of five *C. haemulonii* isolate cases that presented with the susceptibility profile testing method (Appendix A). In these cases, the interpretation results were performed according to the susceptibility breakpoints established by the CLSI: ≤8 mg/L for fluconazole, ≤0.125 mg/L for itraconazole and ketoconazole, ≤4 mg/L for flucytosine, ≤1 mg/L for voriconazole, posaconazole, and amphotericin B, and ≤2 mg/L for caspofungin, anidulafungin, and micafungin [49,51,60]. According to these breakpoints, a total of four *C. haemulonii* isolates had high MICs to amphotericin B, fluconazole, and itraconazole (Appendix A). An isolate from a candidemia case reported in Brazil, with a previous history, showed a high MIC to 5-flucytosine [49]. No *C. haemulonii* isolate showed high MICs to echinocandins.

#### 3.7.3. *C. fermentati* (New Nomenclature: *Pichia fermentans*)

The susceptibility profile results of *C. fermentati* (Appendix A) do not present significant discrepancies. No isolate showed high MICs to azoles such as fluconazole and voriconazole. Itraconazole was the exception since three isolates showed a high MIC to this azole. Only one isolate revealed a high MIC of amphotericin.

Konuma, T. et al. reported one case of *C. fermentati* candidemia in which an *FKS*1 mutation was identified, and, according to the authors, the susceptibility testing results showed a high MIC to echinocandins [59].

A case report of *C. fermentati* published in 2018 demonstrated a discrepancy between the in vitro and in vivo susceptibility to antifungal drugs, particularly in echinocandins [34]. A patient whose isolate revealed low MICs for amphotericin B and micafungin was initially treated with amphotericin and micafungin; however, blood cultures were persistently positive, and, ultimately, the patient died [34].

#### 3.7.4. *C. nivariensis* (New Nomenclature: *Nakaseomyces nivariensis*)

Of the three cases of *C. nivariensis* invasive infection reported in the present work, none of them used CLSI broth microdilution. Cartier N. et al. performed two different instances of susceptibility testing in the same *C. nivariensis* isolate: Etest and broth microdilution according to EUCAST [42]. The results presented in Appendix A showed distinct MICs for azoles (fluconazole, voriconazole, and posaconazole), amphotericin B, and micafungin.

#### 3.7.5. *C. norvegensis* (New Nomenclature: *Pichia norvegensis*)

A few cases have been reported of *C. norvegensis* associated with invasive infection, and most of the cases are associated with acute comorbidities such as solid and liquid malignancy or immunosuppression [25]. In fact, *C. norvegensis* was reported only in three included cases, and all of the patients were immunosuppressed and had leukemia [38,47,68]. The *C. norvengensis* susceptibility profile indicates high MICs for fluconazole in two included cases [38,68]. The echinocandins susceptibility profile was performed in only one case, through caspofungin MICs, which showed low values (0.047 mg/L) [38].

#### 3.7.6. *C. kefyr* (New Nomenclature: *Kluyveromyces marxianus* or *Candida pseudotropicalis*)

In recent years, the number of *C. kefyr* invasive infection reports increased [38]. In our work, four cases were included, and all showed a low MIC to fluconazole and itraconazole [40,46,57,67]. Two cases reported in Europe showed a high MIC to amphotericin [46]. One included case did not present numerical MICs. However, the authors concluded that the isolates were susceptible to amphotericin and azoles [57].

#### 3.7.7. *C. famata* (New Nomenclature: *Debaryomyces hansenii*), *C. bracarensis* (New Nomenclature: *Nakaseomyces bracarensis*), *C. khanbhai*, *C. blankii*, and *C. duobushaemulonii*

The susceptibility testing results of *C. duobushaemulonii* and *C. khanbhai* revealed a high MIC for fluconazole [57,58]. The *C. famata* susceptibility profile was documented in four cases (Appendix A), and the results suggested a susceptibility to both the polyene and azole antifungal classes [34,59]. On the contrary, the *C. blankii* isolate susceptibility profile showed high MICs for both amphotericin and azoles (Appendix A).

The susceptibility profile of the only case of *C. bracarensis* candidemia was performed with the YeastOne method, and results showed low MICs to all antifungals tested (Appendix A) except for itraconazole, whose susceptibility was classified as “susceptible dose-dependent” [53].

## 4. Discussion

Invasive infections by uncommon *Candida* species represent a significant and complex challenge in clinical microbiology and infectious disease management. These uncommon species are of particular concern due to their potential for resistance to standard antifungal treatments and their varying virulence profiles, being a great challenge for patients and clinicians.

### 4.1. Population and Comorbidities

According to numerous studies, the male gender has been identified as a risk factor for ICI, with a prevalence of between 52% and 60% [69,70]. A recent systematic review developed by Egger M. et al. has shown that gender is a risk factor for ICI, where females represent 51.2% of ICI [71]. According to the authors, immune responses can vary between females and males due to a variety of factors, including genetic and hormonal influences. For instance, the presence of two X chromosomes and the hormonal effects of estrogen and progesterone may contribute to different immune response patterns. However, it is important to recognize that immune responses are highly dynamic and context-dependent, and further investigation is mandatory in order to unveil the link between biological gender and ICI [71]. This fact was not in concordance with the studies enrolled in this systematic review. Moreover, there are few case reports of invasive infections by uncommon *Candida* species when compared with the most prevalent *Candida* species.

Other risk factors for ICI beyond demographic factors and gender also include comorbidities and medical interventions [11]. Immunosuppression conditions such as transplantation or HIV, solid and hematological malignancies, corticoid therapy, kidney diseases, and neutropenia are known major risk factors for ICI [11]. The mechanisms involved in each condition have been studied, noticing that the destabilization of the immune system is the theory basis [72,73]. In this review, it is worth noting the high proportion of transplanted and oncology patients. Diabetes was also a common comorbidity in the studies included. It is known that *Candida* spp. are capable of growing in biofilm forms, exhibiting enhanced resistance against most antifungal agents [74]. High levels of glucose in the blood (common in diabetes) promote an increase in *Candida* biofilm formation and, consequently, develop pathogenicity mechanisms [71]. Other important predisposing factors leading to ICI include recent antifungal treatments, previous broad-spectrum antibiotic therapy, and mechanical ventilation [75]. All these risk factors were presented in several patients in this review associated with unfavorable clinical outcomes. 

### 4.2. Uncommon Candida spp. Identification Challenging

It is known that difficulties with *Candida* species identification, especially the emergence of new *Candida* species and the lack of awareness, have resulted in transmission and several outbreaks which have remained unnoticed [76]. The development of rapid and precise methods for identifying uncommon *Candida* invasive agents is a challenge, but it is imperative for patient care and to certify the appropriate implementation of measures to ensure the prevention of infection [22,77,78,79].

Among the conventional methods that traditionally identify pathogenic yeast based on biochemical and morphological characteristics at the species level that are commercially available, the VITEK^®^ 2 and API^®^ AUX systems are the most routinely used systems in clinical microbiology [80]. Unfortunately, the databases of these methods are limited, time-consuming, require high experience in this area, and generally identify only the most common pathogens’ yeasts [80]. In the last few years, evolution in molecular biology has provided the increasing use of DNA techniques as tools for yeast identification and characterization with a high accuracy [81]. Recently, MALDI-TOF and DNA sequencing (ITS and D1/D2 regions) have been discussed as accurate identification methods for *C. auris* [82].

An included study in this systematic review developed by Ruan, S. Y. et al. reported that the VITEK 2 system, in comparison with the VITEK^®^ 1 and API^®^ 32C system, is a better solution to *C. haemulonii* identification [60]. The VITEK^®^ 2 system results were highly consistent with molecular methods for the sequence analysis of the rRNA [60]. In fact, these conclusions are in concordance with the other five included cases in which *C. haemulonii* was correctly identified with the VITEK^®^ 2 system (Appendix A).

The identification of *C. fermentati* in one of three cases was successfully performed with the VITEK^®^ 2 system. DNA sequencing was performed, identifying *C. fermentati* in all the three cases. Konuma, T. and colleagues documented the VITEK^®^ 2 system misidentification of *C. fermentati* in a fungemia case [59]. The authors confirmed that this result is consistent with recent publications: *C. fermentati* is commonly misidentified as *C. famata* by the VITEK^®^ 2 system [83,84,85]. Since it is a cryptic species of *C. guilliermondii*, *C. fermentati* misidentification as *C. guilliermondii* consequently leads to underreported infections due to this pathogenic yeast [34].

*C. nivariensis* identification is another challenge in mycology laboratories since it is routinely misidentified as *C. glabrata* [86]. Poor identification with conventional methods is common. In the present work, all the isolates identified with the VITEK^®^ or API^®^ AUX systems were misidentified, or the identification was not conclusive [39,42]. Fortunately, in order to distinguish *C. nivariensis* from *C. glabrata*, several molecular methods, such as multiplex PCR, a fast, cost-effective, and reliable tool, can be used [87]. Currently, five different genetic Clades have been found related to *C. auris* species [88]. The South-Asian (Clade I), East-Asia (Clade II), Africa (Clade III), and South America (Clade IV) clades, and, more recently, a fifth clade (Clade V), separated from the others by more than 200,000 SNPs, were confirmed to have appeared in patients from Iran [89]. In this review, only four cases had results from a genetic analysis related to clade identification. Clade I was found in three cases reported in Europe (Netherlands, Greece, and Belgium) [41,44,45]. Two patients had a history of hospitalization in Asia, and, in another case (isolated in Greece), they had no travel information, but the authors suggested a horizontal transmission, since *C. auris* with identical sequences was detected from environmental screening.

### 4.3. Susceptibility Interpretation and Treatment Challenges

Antifungal susceptibility testing is an increasingly vital tool for understanding local and worldwide disease epidemiology, considered by clinicians to guide their treatment decision-making process, the treatment of fungal illnesses, and identifying antifungal resistance [90]. Distinct methods are currently available for evaluating the susceptibility of fungi to different antifungals. Broth microdilution is one of the most common susceptibility methods used in clinical and research microbiology laboratories.

Over the years, several commercial susceptibility methods have been available with the advantage of easier interpretation and effective MICs such as Etest and YeasOne [91]. Broth microdilution from CLSI and EUCAST, Etest^®^, and Sensititre^®^ YeastOne were the most used methods in our cases.

Regarding susceptibility testing, there are two different approved reference procedures based on microdilution techniques for antifungal susceptibility testing: CLSI and EUCAST [92,93]. CLSI and EUCAST define breakpoints for *C. albicans* and other common *Candida* non-*albicans*, such as *C. glabrata*, *C. parapsilosis*, and *C. tropicalis* [94]. In 2020, the CDC suggested tentative MIC breakpoints for *C. auris*, and it was applied in the most recent published cases included in this systematic review [41,58,95]. Recently, CLSI also suggested breakpoints for some uncommon *Candida* spp, including species included in the present study, such as *C. haemulonii*, *C. duobushaemulonii*, and *C. kefyr* [96]. Although these new breakpoints are for uncommon species, most of the studies included in the present work emphasize the challenges regarding the interpretation of susceptibility results. Several papers revealed that, due to the absence of clinical breakpoints for the majority of uncommon *Candida* species, MICs obtained have been compared to the susceptibility cutoffs determined for common *Candida* species by CLSI or EUCAST [34,39,47,67]. Nonetheless, it has been reported that susceptibility varies significantly among *Candida* species [59]. Morita k. et al. used CLSI cutoff values of *C. guilliermondii* for the interpretation of susceptibility results of *C. fermentati* isolated from a case of candidemia [34]. Regarding *C. guilliermondii* and *C. fermentati* susceptibility profiles, there seems to be not much difference, with echinocandins having good antifungal activity against both species [34,97]. On the other hand, *C. fermentati* micafungin resistance has been documented and is associated with *FKS*1 mutation [59].

Multidrug-resistant *Candida* spp species are increasingly reported [97]. It was also observed in the present work that several isolates of distinct *Candida* spp. had high MICs for azoles, polyenes, and echinocandins, mainly fluconazole. *Candida* spp. antifungal resistance mechanisms can be conferred through gain-of-function mutations in different target pathway genes or in their transcriptional regulators [98]. The present work included four *C. auris* and *C. fermentati* cases, which analyzed the presence of the most known mutations associated with a reduced susceptibility to fluconazole and echinocandins: ergosterol biosynthesis pathways mutations (*ERG*11) and cell wall biosynthesis mutation (*FKS*1), respectively [32,58,59,65].

### 4.4. Global Epidemiology

Asia was the epicenter of several uncommon *Candida* species [20,99]. It was noted that there was a high incidence of reported cases documenting high MICs for distinct classes of antifungal agents (Appendix A). Due to this emergence of uncommon *Candida* species antifungal resistance, susceptibility testing is crucial to guide the management of ICI in Asian countries [35]. *C. khanbhai* associated with invasive infection was reported for the first time in Malaysia recently with a high MIC to amphotericin and azoles in a patient with no comorbidities, who died after hospitalization due to new-onset hospital-acquired pneumonia and posterior diagnosis of candidemia associated with this new *Candida* spp. 

Asia is one of the epicenters of antimicrobial drug resistance and a growing concern related to the dissemination of multidrug-resistant pathogens [100]. A meta-analysis conducted by Habibzadeh, A. et al. compared fungal drug resistance between Europe and Asia [99]. The results demonstrated that Asia had a high prevalence, which was justified due to the poor global health infrastructure in most Asian countries [99]. These findings may be one of the reasons why Asia was the continent with the highest number of reported cases in our work.

Europe presented the highest number of *C. auris* isolates with a high MIC to fluconazole and voriconazole. Although the reasons for the changing epidemiology of invasive fungal infections in Europe are not entirely known, Lass-Flörl and colleagues suggested that it probably results from different factors, such as the increased use of fluconazole prophylaxis in cancer and transplant patients and changes in treatment strategies for various at-risk populations [101].

In Europe, mortality associated with *C. auris* invasive infections was reported [37,41,48]. One of the most puzzling characteristics of *C. auris* is the recent simultaneous and independent emergence of five genetically distinct clades on three continents [102]. It is known that the thermotolerance of *C. auris* compared with other *Candida* species is phylogenetically similar. Indeed, it has been suggested that global warming is a contributing factor to the emergence of *C. auris* [102]. Although the association between human and animal health and the shared environment has been documented in recent years, the current understanding of the effects of climate change and risk for yeast diseases is still incompletely understood [103]. However, Bajpai, Vivetek K. et al. documented the change in climate as a major reason for the growth of a number of these invasive infections, suggesting that global warming and moisture effects on pathogen sporulation and dispersal could benefit certain pathogens and the introduction of new potential vectors [104].

The European Centre for Disease Prevention and Control conducted surveys related to epidemiology, laboratory capacity, and preparedness for *C. auris* in Europe in two distinct periods: 2013 to 2017, and January 2018 to May 2019 [105]. However, since the information was not updated after the COVID-19 pandemic, a *C. auris* survey was conducted in April 2022 in order to understand the epidemiological situation and control measures implemented regarding *C. auris* infections in Europe [106]. According to the obtained data, *C. auris* was assessed as endemic in Spain, which was one of the first countries to report outbreaks, and five countries reported outbreaks in the period of 2019 to 2021: France, Greece, Germany, Denmark, and Italy [106]. Only two *C. auris* cases were not reported from outbreak zones: Belgium and the Netherlands [44,45].

In European countries, the distribution of invasive infections by uncommon *Candida* species differs from one country to another. *C. norvegensis* was reported in two cases, both transplanted patients [38,47]. This pathogen was isolated for the first time in asthmatic patients in 1954, in Norway [38]. In 1990, the first case of *C. norvegensis* invasive infection was reported in Denmark [107]. According to Pfaller M. et al., *C. norvegensis* is more prominent in Eastern European [108]. This fact concurs with the results of the present work since there are only three cases of *C. norvegensis*, two of which were reported in Europe [38]. In both cases, the patients underwent liver transplantation, a risk factor associated with *C. norvegensis* infection [25].

*C. nivariensis* was first described in Spain in 2005, collected from one institution from bronchoalveolar lavage, blood culture, and urine samples [109]. Only two cases of *C. nivariensis* invasive infections in Europe were included [39]. In contrast, no *C. bracarensis* cases were identified. In these cases, the importance of optimizing the identification techniques available is highlighted, focusing on speed, simplicity, and reliable results, suggesting MALDI-TOF MS as a suitable option [39]. The reported *C. nivariensis* candidemia in Spain was associated with nosocomial infection [39]. The authors documented the gardens and flowers presented in the hospital as possible sources of contamination [39].

*Candida nivariensis* and *C. bracarensis* are two species related to *C. glabrata*, being phenotypically indistinguishable [109,110,111].

*Candida kefyr* was reported in Europe in two case reports [40,46]. This species has gained recent importance as an emerging opportunistic yeast predominantly in immunocompromised patients [40]. *Candida kefyr* pyelonephritis cases are limited [40]. The authors highlighted the possibility of bloodstream dissemination from the urinary system or hematogenous spread to the kidneys.

Evidence from the included cases from American countries suggests that, although uncommon species are rare, they are becoming pathogens, namely, in immunosuppressed patients with high mortality rates [49,112]. In several studies, the challenges associated with the lack of economic, rapid, and accurate identification and susceptibility methods were also reported [50,65]. Although, in this systematic review, the number of cases in South America is limited, Sifuentes-Osornio, J. and colleagues documented that the incidence of candidemia is higher in this region than in Europe and the United States [113]. The authors also reported that the *Candida* spp. distribution in Latin America is distinct from other regions, which was associated with huge variations in the quality of health care, namely, in high-risk patients. In fact, *C. haemulonii* was the pathogen most reported in this region, in contrast to all the other locations.

With the present work, it was possible to understand the concordance in the challenges related to the emergence of uncommon invasive *Candida* species. It was also possible to understand that some species are most related to specific comorbidities; for example, in the included cases, *C. norvegensis* formed the majority of the species present in transplanted patients [38,47]. It was also observed that there was resistance to fluconazole of distinct uncommon *Candida* species independently of geographical localization [56].

In 2016, Calvo B., et al. reported the first outbreak of *C. auris* in America [114]. In 2018, an epidemiological survey in the USA reported that all the isolates were genetically related to the South American, African, East Asian, or South Asian clades, suggesting that *C. auris* was introduced in the USA through travel-related cases [115].

Global change, encompassing climate shifts, increased international travel, and altered land use, has promoted shifts in the natural balance between humans and micro-organisms, favoring the emergence of novel, resistant strains that challenge current medical therapies and public health initiatives.

### 4.5. Limitations

There are several studies reporting ICI due to uncommon *Candida* species; however, the exclusion of studies with no information regarding the clinical outcome or susceptibility results may have led to a bias in the selection and results.

Our findings are limited, nevertheless, by the quality and breadth of the low number of cases and data in the reports, which were not uniform (for example, distinct identification and susceptibility methods; and different comorbidities). The low number of studies related to some of the uncommon species may limit us from reaching conclusions.

## 5. Conclusions

It is undeniable that, in recent years, uncommon *Candida* species have emerged worldwide. Over the last few years, numerous hospitals have reported a significant and progressive shift in the etiology of invasive candidiasis in diverse patient populations and distinct hospital settings [116]. In our work, diagnostic delays and aggravating related to difficulty in fast and accurate identification were reported in the vast majority of the included studies [37,42]. Early and accurate identification and tailored antifungal therapy are essential for effective management and improving patient outcomes. Ongoing research into the pathogenesis, diagnosis, and treatment of these infections is critical to address the challenges they present in the clinical setting worldwide.

## Figures and Tables

**Figure 1 jof-10-00558-f001:**
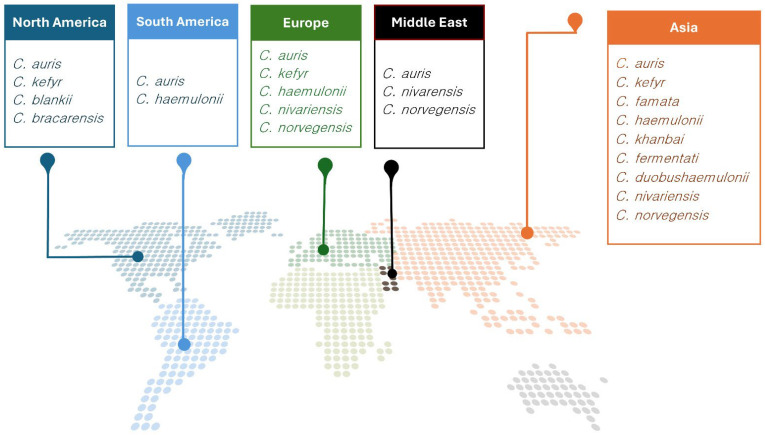
Global epidemiology of uncommon *Candida* spp. invasive infections.

**Figure 2 jof-10-00558-f002:**
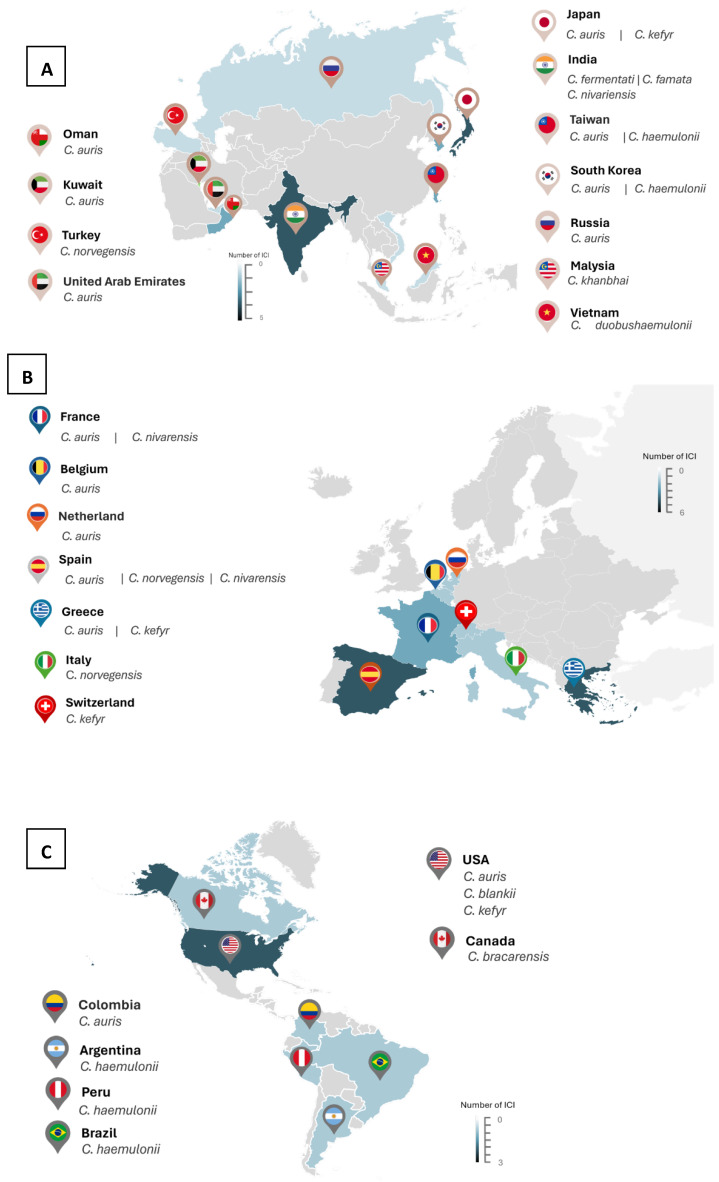
Epidemiology of uncommon *Candida* species invasive infections in each continent: (**A**) Asia; (**B**) Europe; and (**C**) America. Invasive *Candida* Infection (ICI).

## Data Availability

Not applicable.

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
