# Peer review of "Global Epidemiology of Invasive Infections by Uncommon Candida Species: A Systematic Review"

_jof, 2024, doi:10.3390/jof10080558_

Round 1
Reviewer 1 Report
The manuscript is an example of a well-written, very interesting and comprehensive review. The manuscript describes currently known cases of invasive infections caused by members of the genus Candida, with the exception of the well-known C. albicans, C. glabrata, etc., the so-called non-spreading Candida species, which are a growing medical problem. I have not found any serious shortcomings, so there are only minor comments.
1. In some places, species and genus names are not written in italics, which should be corrected. For example:
· Str. 97-99, “C. albicans, C. glabrata, ..., C. dubliniensis”
· Str. 488, “C. glabrata”
· Str. 491, “C. glabrata”
· Str. 515, “C. glabrata, C. parapsilosis, and C. tropicalis”
· Str. 527-528, “C. guilliermondii”
· Str. 604, “nivariensis” and “C. glabrata”
There may be more such places. Such typos need to be carefully checked for and corrected.
2. In the Materials and Methods section, it should be separately clarified what the "inclusion criteria" is and how it was applied.
3. This may be related to the previous point, but it is necessary to clarify on what principle 825 publications were excluded from the analysis.
4. This point is left to the authors' discretion, but it seems to me that it would be more informative if figures (especially Figure S3) were placed in the main text and tables in the Supplementary Materials. Figure S3 could even be a graphical summary of the article.
5. On the map (Figure 1), the Middle East should be drawn in dark color. The information on North and South America should be separated. In the Asia section, "Auris" should be lowercase.
6. If it is known which mutations in the FKS1 gene occur and how they affect its activity, this should be added.
7. In the section where C. auris resistance is described, it should be added which antifungals have the lowest MICs, which can be the basis for future treatment strategies.
8. "Invasive," "deep-seated," and other similar words on str. 82-84 should be enclosed in quotes.
9. According to str. 220-221, “The age range was from 26 to 88 years, with a mean and standard deviation of 59,1±16.6 years”. But according to str. 264-265, “The average age of the population who ended up dying was 59,4±16,6 years (range 26-88 years)”. This should probably be corrected if it's an accidental typo.
Author Response
Responses to Reviewer 1 Comments to the Authors:
Major comments
The manuscript is an example of a well-written, very interesting and comprehensive review. The manuscript describes currently known cases of invasive infections caused by members of the genus Candida, with the exception of the well-known C. albicans, C. glabrata, etc., the so-called non-spreading Candida species, which are a growing medical problem. I have not found any serious shortcomings, so there are only minor comments.
The authors would like to thank the Reviewer for the insightful feedback and valuable suggestions that have greatly improved our paper. Please find the point-by-point responses below. Changes in the manuscript are marked in yellow.
Detail comments
- In some places, species and genus names are not written in italics, which should be corrected. For example:
- Str. 97-99, “C. albicans, C. glabrata, ..., C. dubliniensis”
- Str. 488, “C. glabrata”
- Str. 491, “C. glabrata”
- Str. 515, “C. glabrata, C. parapsilosis, and C. tropicalis”
- Str. 527-528, “C. guilliermondii”
- Str. 604, “nivariensis” and “C. glabrata”
There may be more such places. Such typos need to be carefully checked for and corrected.
R: Changes were made as suggested.
- In the Materials and Methods section, it should be separately clarified what the "inclusion criteria" is and how it was applied. 3. This may be related to the previous point, but it is necessary to clarify on what principle 825 publications were excluded from the analysis.
R: Detailed information has been added to the inclusion and exclusion criteria (lines 100-113).
- This point is left to the authors' discretion, but it seems to me that it would be more informative if figures (especially Figure S3) were placed in the main text and tables in the Supplementary Materials. Figure S3 could even be a graphical summary of the article.
R: Changes were made as suggested. Table 1 and 2 moved to supplementary material, Table S4 and S5, respectively. Figure S3 is now on the main text as “figure 2”.
- On the map (Figure 1), the Middle East should be drawn in dark color. The information on North and South America should be separated. In the Asia section, "Auris" should be lowercase.
R: Changes were made as suggested.
- If it is known which mutations in the FKS1 gene occur and how they affect its activity, this should be added.
R: Information regarding mutations in FKS1 and ERG11 genes has been added (Table S5). All FKS1 mutations occurred in hotspot 1. This information has been added to the text (line 362).
- In the section where C. auris resistance is described, it should be added which antifungals have the lowest MICs, which can be the basis for future treatment strategies.
R: Information has been added to the C. auris section (lines 356 and 357).
- "Invasive," "deep-seated," and other similar words on str. 82-84 should be enclosed in quotes.
R: Changes were made as suggested.
- According to str. 220-221, “The age range was from 26 to 88 years, with a mean and standard deviation of 59,1±16.6 years”. But according to str. 264-265, “The average age of the population who ended up dying was 59,4±16,6 years (range 26-88 years)”. This should probably be corrected if it's an accidental typo.
R: The first age range refers to all the patients with invasive infection enrolled in the selected studies. The average range of 59,4±16,6 refers only to patients with poor outcome (death).
Reviewer 2 Report
I received the manuscript no. jof-3109677 (Rising Threat: A Systematic Review of the Global Epidemiology of Invasive Infections by Uncommon Candida Species) to review on June 2024.
That´s an interesting proposal of presenting the importance of uncommon Candida species causing invasive infection in the last 2 to 3 decades. However, it is not clear the criteria to classify uncommon species. The methods must be reviewed. There are also lots issues in presenting references.
The twenty-three pages of this manuscript needs to be reviewed. The 2 tables are too big and don´t bring too much information. It is not the authors fault, there is lack of standardization of methods (identification and AFST) and the way to present it must be revisited.
Please find more information in the attached document (PDF)

Author Response
Responses to Reviewer 2 Comments to the Authors:
Major comments
The manuscript "Rising Threat: A Systematic Review of the Global Epidemiology of Invasive Infections by Uncommon Candida Species" is an excellently written and conducted paper. It contains noteworthy information about the epidemiology, diagnosis, susceptibility profiles, and clinical presentation outcomes of rare Candida species responsible for invasive infections. However, there are some suggested comments to be considered.
The authors would like to thank the Reviewer for the insightful feedback and valuable suggestions that have greatly improved our paper. Please find the point-by-point responses below. Changes in the manuscript are marked in yellow.
In the introduction, it's important to mention the current taxonomy of non-Candida albicans species. I also think it's important to mention that C. auris is currently divided into four main clades that show different biological patterns and drug resistance: I, from South Asia; II, from East Asia; III, from South Africa; IV, from South America. Recently, clade V has been identified in Iran, as the manuscript mentions that isolates from Asia and Europe were closely related to isolates from different clades.
R: Information regarding the current taxonomy has been added. Information regarding the C. auris Clades has been added in the discussion section (lines 500-502).
Line 436. In the manuscript, it is mentioned “although according to authors, the immune response in females is stronger than in men”, explain in what sense the immune response is stronger in women than in men, and how hormonal factors contribute.
R: Information regarding immune response in females has been added (lines 441-444).
Detail comments
Lines 52 and 54. Change “Candida albicans” to “C. albicans”
Line 62. Change “spp to “spp.”
Lines 97 and 98. Put in italics “C. albicans, C. glabrata, C. parapsilosis, C. tropicalis, C. lusitaniae, C. guilliermondii, C. krusei and C. dubliniensis.”
Lines 197 and 200. Figures 3b and 3c change to Figures S3b and S3c.
References must be carefully reviewed and strictly follow the format requested by the journal.
R: All changes have been made as suggested and are highlighted in the main document. All references were carefully reviewed and formatted.
Reviewer 3 Report
The manuscript "Rising Threat: A Systematic Review of the Global Epidemiology of Invasive Infections by Uncommon Candida Species" is an excellently written and conducted paper. It contains noteworthy information about the epidemiology, diagnosis, susceptibility profiles, and clinical presentation outcomes of rare Candida species responsible for invasive infections. However, there are some suggested comments to be considered.
In the introduction, it's important to mention the current taxonomy of non-Candida albicans species. I also think it's important to mention that C. auris is currently divided into four main clades that show different biological patterns and drug resistance: I, from South Asia; II, from East Asia; III, from South Africa; IV, from South America. Recently, clade V has been identified in Iran, as the manuscript mentions that isolates from Asia and Europe were closely related to isolates from different clades.
Line 436. In the manuscript, it is mentioned “although according to authors, the immune response in females is stronger than in men”, explain in what sense the immune response is stronger in women than in men, and how hormonal factors contribute.
Lines 52 and 54. Change “Candida albicans” to “C. albicans”
Line 62. Change “spp to “spp.”
Lines 97 and 98. Put in italics “C. albicans, C. glabrata, C. parapsilosis, C. tropicalis, C. lusitaniae, C. guilliermondii, C. krusei and C. dubliniensis.”
Lines 197 and 200. Figures 3b and 3c change to Figures S3b and S3c.
References must be carefully reviewed and strictly follow the format requested by the journal.
Author Response
Responses to Reviewer 3 Comments to the Authors:
I received the manuscript no. jof-3109677 (Rising Threat: A Systematic Review of the Global Epidemiology of Invasive Infections by Uncommon Candida Species) to review on June 2024.
That´s an interesting proposal of presenting the importance of uncommon Candida species causing invasive infection in the last 2 to 3 decades. However, it is not clear the criteria to classify uncommon species. The methods must be reviewed. There are also lots issues in presenting references.
The twenty-three pages of this manuscript needs to be reviewed. The 2 tables are too big and don´t bring too much information. It is not the authors fault, there is lack of standardization of methods (identification and AFST) and the way to present it must be revisited.
The authors would like to thank the Reviewer for the insightful feedback and valuable suggestions that have greatly improved our paper. Please find the point-by-point responses in the pdf document in the attachment. Changes in the manuscript are marked in yellow.
